# Circulating Neurofilament Light Chain Levels Increase with Age and Are Associated with Worse Physical Function and Body Composition in Men but Not in Women

**DOI:** 10.3390/ijms241612751

**Published:** 2023-08-13

**Authors:** Xavier Capo, Aina Maria Galmes-Panades, Cayetano Navas-Enamorado, Ana Ortega-Moral, Silvia Marín, Marta Cascante, Andrés Sánchez-Polo, Luis Masmiquel, Margalida Torrens-Mas, Marta Gonzalez-Freire

**Affiliations:** 1Translational Research in Aging and Longevity (TRIAL) Group, Health Research Institute of the Balearic Islands (IdISBa), 07120 Palma de Mallorca, Spain; xaviercapofiol@hotmail.com (X.C.); aina.galmes.panades@gmail.com (A.M.G.-P.); caye.navas.enamorado@gmail.com (C.N.-E.); aortegam1695@hotmail.com (A.O.-M.); poloasp@gmail.com (A.S.-P.); lida.torrens@gmail.com (M.T.-M.); 2Physical Activity and Sport Sciences Research Group (GICAFE), Institute for Educational Research and Innovation (IRIE), University of the Balearic Islands, 07120 Palma de Mallorca, Spain; 3Consorcio CIBER, M.P. Fisiopatología de la Obesidad y Nutrición (CIBERObn), Instituto de Salud Carlos III (ISCIII), 28029 Madrid, Spain; 4Department of Biochemistry and Molecular Biomedicine, Faculty of Biology, Universitat de Barcelona, 08028 Barcelona, Spain; silviamarin@ub.edu (S.M.); martacascante@ub.edu (M.C.); 5Institute of Biomedicine of University of Barcelona (IBUB), University of Barcelona, 08028 Barcelona, Spain; 6CIBEREHD, Network Center for Hepatic and Digestive Diseases, National Spanish Health Institute Carlos III (ISCIII), 28029 Madrid, Spain; 7Vascular and Metabolic Pathologies Group, Health Research Institute of the Balearic Islands (IdISBa), 07120 Palma de Mallorca, Spain; lmasmiquel@gmail.com; 8Grupo Multidisciplinar de Oncología Traslacional, Institut Universitari d´Investigació en Ciències de la Salut (IUNICS), University of the Balearic Islands, 07122 Palma de Mallorca, Spain; 9Faculty of Experimental Sciences, Francisco de Vitoria University (UFV), 28223 Madrid, Spain

**Keywords:** aging, neurodegeneration, metabolomics, NFL, muscle function, gender dimorphism

## Abstract

This study aimed to assess the relationship between age-related changes in Neurofilament Light Chain (NFL), a marker of neuronal function, and various factors including muscle function, body composition, and metabolomic markers. The study included 40 participants, aged 20 to 85 years. NFL levels were measured, and muscle function, body composition, and metabolomic markers were assessed. NFL levels increased significantly with age, particularly in men. Negative correlations were found between NFL levels and measures of muscle function, such as grip strength, walking speed, and chair test performance, indicating a decline in muscle performance with increasing NFL. These associations were more pronounced in men. NFL levels also negatively correlated with muscle quality in men, as measured by 50 kHz phase angle. In terms of body composition, NFL was positively correlated with markers of fat mass and negatively correlated with markers of muscle mass, predominantly in men. Metabolomic analysis revealed significant associations between NFL levels and specific metabolites, with gender-dependent relationships observed. This study provides insights into the relationship between circulating serum NFL, muscle function, and aging. Our findings hint at circulating NFL as a potential early marker of age-associated neurodegenerative processes, especially in men.

## 1. Introduction

Aging is an inevitable, natural, and irreversible process that occurs as we get older, and it is characterized by a variety of physical, cognitive, and social changes [1]. Aging is a significant risk factor for the development of neurodegenerative diseases such as Alzheimer’s disease (AD) and Parkinson’s disease (PD) among others.

Traditionally, tests such as grip strength, 4-m gait speed (at a usual or a fast pace), or the chair test have been used as markers or indicators of neurological and muscle function [2,3,4]. In this regard, compelling evidence indicates a robust relationship between grip strength and neurological function, particularly concerning brain health and cognitive functioning in older adults. [3,4,5]. Additionally, age-related decline in grip strength has been linked to the functional connectivity of brain regions involved in the generation of dynamic grip [6]. The 4-m gait speed test, conducted at both usual and fast pace, is linked to higher levels of cognitive decline in elderly individuals. These tests indicate an elevated risk of future cognitive decline and the development of dementia [2,3,4]. Both grip strength and gait speed are gold standards for measuring frailty and decline in muscle function with aging and are predictive of mortality [5].

Several plasma markers such as Tau protein, neurofilament light chain (NFL), plasma amyloid beta, and ptau181, among others, can be used to track the progression of neurodegenerative diseases like AD or PD [7,8]. NFL is a neuronal protein that constitutes a portion of the neurofilament cytoskeleton and is primarily present in the axons. It plays a crucial role in determining the axonal diameter. NFL can be detected in both cerebrospinal fluid and blood, emerging as a potential biomarker for ongoing axonal compromise. [9,10]. In this sense, some studies have shown that blood levels of NFL increase with age and are associated with changes in cognitive function [11,12]. NFL has been proposed as a potential biomarker for various neurodegenerative diseases, such as AD, PD, multiple sclerosis, amyotrophic lateral sclerosis (ALS), and traumatic brain injury [13,14,15]. NFL has also been investigated as a potential marker for aging and age-related neuronal changes. As individuals age, there is a natural decline in neuronal health and function and age-related changes can affect the integrity of the nervous system [16,17]. Previous studies have evidenced a progressive increase in NFL plasma levels with age, this increase is probably associated with normal brain atrophy that occurs with aging [9,18]. However, it is important to note that the increase in NFL levels with age is relatively modest compared to the elevated levels observed in neurodegenerative diseases [19]. Higher NFL levels in older individuals have been associated with various factors, including cognitive decline, poor physical performance, and an increased risk of age-related neurological conditions [15,19,20]. It has also been proposed that the appearance of age-related comorbidities can affect the increase in plasma NFL levels with aging [9,18]. In healthy populations, NFL plasma levels can be influenced by renal function, body composition, age, and blood volume [16,21]; however, some authors have suggested that the impact of these factors on NFL levels is attenuated after the age of 60 [19]. There is growing evidence that changes in metabolic pathways may contribute to the development and progression of neurodegenerative diseases [22]. In this sense, changes in plasma and brain metabolomic profiles have been observed in several neurodegenerative diseases such as AD, PD, and normal aging [23,24,25].

Dysregulation in lipid metabolism may contribute to neuroinflammation, oxidative stress, decline in physical function with aging, and the formation of pathological protein aggregates [26]. Plasma metabolomic analysis has shown perturbations in amino acid levels, particularly alterations in aromatic amino acids (such as tryptophan and phenylalanine), branched-chain amino acids (BCAAs) (e.g., leucine, isoleucine, and valine), and lysophosphatidylcholines (LPC), with aging and in individuals with neurodegenerative disorders [27,28,29,30,31,32]. In association with metabolomic changes, it is also evidenced that neurodegeneration and aging can also induce changes in body composition which in turn might accelerate the onset of neurodegenerative processes and or diseases [33,34].

The aim of this study was to evaluate changes in NFL with age, and to evaluate its relationship with known proxies of muscle function such as grip strength, chair test, and walking speed in a healthy cohort of adults from the Balearic region of Mallorca. We hypothesized that circulating NFL increases with aging, and this increase would be associated with a decline in physical function, as well as with changes in metabolomic markers.

## 2. Results

### 2.1. Participant Characteristics

The demographic and clinical characteristics of the 40 participants of the study are shown in Table 1. The overview of the study is shown in Figure 1.

The study included participants with an average age of 47.8 ± 2.7 years, (20 years the youngest patient and 85 years the oldest). All patients were Caucasians. The gender distribution of the study population was balanced with 50% of participants in each gender. There were no significant differences in age, sex, or comorbidities. BMI was higher in men compared to women (*p* = 0.02). Gait speed and grip strength were different among sex, with men presenting better physical performance, compared to women (*p* = 0.027 and *p* < 0.001, respectively). In relation to body composition, women presented higher fat levels while men presented higher levels of muscle mass (*p* < 0.001).

The levels of NFL in serum were similar in both men and women (14.0 ± 1.54 and 15.3 ± 1.69, respectively) (Figure 2A). NFL levels increased with age (r = 0.553, *p* < 0.001) (Figure 2B) but when analyzed by gender, men presented a higher correlation with age (r = 0.824, *p* < 0.001), while this association with age disappeared in women (r = 0.281, *p* = 0.230) (Figure 2C).

### 2.2. Circulating NFL and Its Association with Body Composition

The relationship between serum NFL levels and body composition parameters is shown in Table 2. We found a positive and significant correlation between NFL plasma levels and fat percentage (r = 0.337, *p* = 0.034) and visceral fat area (r = 0.319, *p* = 0.045). Muscle quality, measured with inbody 50 kH phase angle, showed a negative correlation with NFL plasma levels (r = −0.406, *p* = 0.009). When considering the gender effect, we found that body fat mass (r = 0.482, *p* = 0.031), fat percentage (r = 0.447, *p* = 0.048), visceral fat area (r = 0.516, *p* = 0.02), and inbody 50 kH phase angle (r = −0.603, *p* = 0.005) were correlated with NFL levels only in men. Unexpectedly, we did not find association between NFL levels and levels of muscle mass ( *p* > 0.05) or creatinine, a surrogate marker of muscle mass (Appendix A).

### 2.3. Relationship between Circulating NFL and Functional Performance Tests

The association between serum NFL levels and functional performance tests is presented in Table 2. Circulating NFL was only associated with the time to perform the chair stand test (r = 0.509, *p* = 0.01). However, when we analyzed the data taking into account the gender effect, we found that in men, NFL levels were negatively correlated with normal walking speed (r = −0.460, *p* = 0.041) and grip strength (r = −0.535, *p* = 0.015), and positively correlated with the chair stand test (r = 0.478, *p* = 0.033) (Figure 3). No correlations were found between circulating NFL and any functional performance tests in women (*p* > 0.05). To account for body weight as a potential confounder, we normalized the grip strength and chair test data by body weight. Interestingly, this normalization had minimal impact on the results, except for grip strength, which showed a significant negative correlation with NFL levels (r = −0.334, *p* = 0.035) (see Appendix A).

### 2.4. Relationship between Circulating NFL and Metabolomic Markers

We tested whether the age-related increase in serum NFL was accompanied by changes in metabolomic markers (Figure 3). In order to do so, we analyzed the circulating metabolites with the AbsoluteIDQ^®^ p180 kit (Biocrates). Correlations between NFL levels and the most significant metabolomic markers are presented in Table 2 and Figure 4. We found that NFL levels were positively correlated with Aspartate (ASP) (r = 0.371, *p* = 0.018), putrescine (r = 0.428, *p* = 0.006), Taurine (TAU) (r = 0.317, *p* = 0.046), Kynurenine (KYN) (r = 0.319, *p* = 0.045), PC_aa_C36:5 (r = 0.395, *p* = 0.013), PC_ae_C40:3 (r = 0.322, *p* = 0.046), PC_ae_C42:3 (r = 0.349, *p* = 0.030) PC_ae_C44:4 (r = 0.457, *p* = 0.003), acetylcarnitine (acylcarnitine C2) (r = 0.325, *p* = 0.041), and OH-Sphingomyelin C22:2 (r = 0.318, *p* = 0.045). However, when we analyzed the data taking into account the gender effect, we found a positive correlation between NFL and Glutamate (GLU) (r = 0.445, *p* = 0.05), PC_aa_C36:5 (r = 0.564, *p* = 0.010), PC_aa_C36:6 (r = 0.460, *p* = 0.041), PC_aa_C38:6 (r = 0.513, *p* = 0.021), PC_aa_C42:5 (r = 0.465, *p* = 0.040), PC_ae_C38:0 (r = 0.477, *p* = 0.034), and PC_ae_C44:4 (r = 0.490, *p* = 0.028), and a negative correlation between NFL and Glycine (GLY) (r = −0.608, *p* = 0.004), Serine (SER) (r = −0.443, *p* = 0.050), and LysoPC_aa_C18:2 (r = −0.458, *p* = 0.042) just in men. In women, we found a positive correlation between ASP (r = 0.446, *p* = 0.049), Putrescine (r = 0.463, *p* = 0.040), LysoPC_aa_C17:0 (r = 0.492, *p* = 0.028), and PC_ae_C42:3 (r = 0.579, *p* = 0.009).

Considering that the correlations between NFL and these metabolites could be influenced by muscle quality, we conducted multivariate linear regression analyses adjusting for age, gender, and muscle quality. Consequently, some of the associations observed initially disappeared, however, others, including Acetylcarnitine (C2) and PC_ae_C40.3, remained significant. The complete set of beta coefficients and corresponding *p*-values for NFL and the metabolites can be found in Appendix A.

## 3. Discussion

In this study, we tested the hypothesis that circulating NFL increased with aging and this increase would be associated with a decline in muscle function and changes in body composition and metabolism. Indeed, we found that serum NFL levels increased with age, and negatively correlated with measures of muscle function. Furthermore, these associations were significantly stronger in men, consistent with previous studies [16,19,35,36]. These findings hint at circulating NFL as a potential early marker of age-associated neurodegenerative processes, especially in men.

It is largely described that functional tests such as grip strength, 4 m gait speed at a usual or a fast pace, or chair test can be good indicators of neurological function [2,3,4]. Although it was not the scope of the study, we found a decline in muscle function with aging, in our cohort, similar to what has been shown in previous studies [37,38]. This decline in muscle performance was strongly associated with an increase in circulating NFL. More specifically, in this sense we have observed a negative correlation between grip strength and walking speed at usual pace and NFL levels in men. These results are in accordance with a previous study in patients with sarcopenia or with muscle mass loss [39]. Some authors have reported a negative correlation between NFL plasma levels and gait speed in women [40], and also in patients older than 60 years [12]. Grip strength is closely tied to neuromuscular function as it involves the coordination between the nervous system and the muscles responsible for gripping [41]. This fact would explain that higher values of NFL would be associated with lower grip strength as we age. These results suggest the use of circulating NFL as a good marker of neuromuscular function in healthy individuals with aging. Similarly, we also observed an association between the time to sit and stand test or chair test and NFL levels, but this correlation was only significant in men. Previous studies have reported that high plasma NFL levels were associated with an increased risk of mobility decline, including chair-stand test performance, in older adults [20,40]. 

Both grip strength and chair tests are directly related to muscle quality [42,43]. In this sense, we found a negative correlation between NFL levels and muscle quality (measured as 50 kHz phase angle) in men consistent with the association between NFL levels and grip strength and the chair test. Changes in muscle quality are normally accompanied by changes in body composition with aging. In our study, we evidenced that serum NFL levels were positively correlated with markers of fat mass (such as fat percentage, visceral fat area, and body fat mass) and negatively correlated with markers of muscle mass. In both cases, these findings were mainly observed in men, but not in women. Contrary to our results, previous studies reported a negative correlation between serum NFL levels and BMI, waist–hip ratio, body cell mass, body fat mass, and total body water [44]. 

There is increasing evidence to suggest that alterations in the metabolome may be involved in the development and progression of neurodegenerative disorders [22,23]. The relationship between metabolomic changes and NFL levels is complex and multifactorial since it may vary depending on the underlying neurodegenerative condition and individual patient characteristics. Explaining these relationships is challenging. Lower or higher circulating levels may stem from a decrease or increase in the number of these metabolites within the cells, leading to reduced or increased release. Furthermore, it could be attributed to the cells requiring more or less of these metabolites, consequently resulting in decreased release or increased uptake. In this study, using targeted metabolomics, we found that some specific metabolites showed a strong association with NFL levels, and these relationships seemed to have a gender dimorphism. Among them, we evidenced a negative correlation between serine and glycine and NFL levels only in men. Serine metabolism is known to play a crucial role in various biological processes, including neuronal development, neurotransmitter synthesis, and myelin formation. Changes in serine metabolism may potentially affect neurodegenerative processes and NFL levels indirectly. In this sense, it is evidenced that L-serine is the precursor of D-serine, and some authors have postulated this as an early marker of AD. Glycine is involved in the development and maturation of synapses; in fact, during the early stages of neural development, glycine receptors contribute to synapse formation and refinement [45]. The fact that these two amino acids are key to the development and establishment of neuronal synapses and the transmission of nerve impulses might explain the negative correlation between the levels of NFL and these amino acids. Aspartate and NFL levels were positively correlated in our cohort, but when analyzed by gender, this relationship only remained significant in women. Aspartate is an excitatory neurotransmitter that plays a key role in neuronal function, particularly through the activation of N-methyl-D-aspartate (NMDA) receptors. However, excessive activation of NMDA receptors by high levels of aspartate can lead to excitotoxicity, a process that results in neural damage and death [46]. This fact could explain the direct correlation between aspartate plasma levels and a neurological damage marker such as NFL. Similarly, glutamate is a critical neurotransmitter in the brain that plays a central role in various aspects of neurological function. A disbalance in its metabolism is associated with neurological disorders, such as stroke, traumatic brain injury, and neurodegenerative diseases like Alzheimer’s and Parkinson’s diseases [47], which could explain the direct correlation between glutamate plasma levels and NFL in men. Finally, it is remarkable that we found a negative correlation between the levels of lysophosphatidylcholine 18:2 and circulating NFL in men. This finding is in line with a previous study from our group, were we showed that low levels of lysophosphatidylcholine 18:2 were an independent predictor of decline in gait speed in older adults [31]. Therefore, these results provide robustness to our findings, and we could hypothesize that changes in NFL might be an earlier indicator of changes in neuromuscular function. However, upon adjusting for muscle quality, the majority of these associations became non-significant, with the exception of acetylcarnitine. Acetylcarnitine is a crucial metabolite involved in fatty acid metabolism and energy production. While limited direct evidence exists on the association between NFL and acetylcarnitine specifically in skeletal muscle, studies have identified alterations in acetylcarnitine levels in various disease states, including neurodegenerative conditions [48,49].

Further replication and confirmation in a larger population are needed to strengthen the validity of our study’s findings. Our study provides compelling evidence that NFL may play a significant role in muscle function with aging, potentially exhibiting gender-specific effects. We believe our results might suggest that circulating NFL could serve as a promising early indicator of age-associated neurodegenerative processes. Further research is needed to better understand the relationship between NFL levels and the aging process, and to determine how NFL measurements can be effectively implemented in the clinic to assess early changes in age-related neurological processes and predict health span outcomes.

## 4. Materials and Methods

### 4.1. Study Subjects

The study included a total of 40 participants, between 20 and 85 years, who were part of an ongoing study: the Balearic Islands Study of Aging (BILSA). Participants included in this study were healthy, active, and free of major diseases, except controlled hypertension or hypercholesterolemia. Participants were excluded if they were smokers or had cognitive impairment, neuromuscular or muscular disorders, and/or history of cancer in the previous 10 years. Participants were specifically instructed to abstain from exercise 24 h prior to the study to prevent any elevation of inflammatory markers in the blood that could confuse or impact the results. The data collection for this study took place between February 2022 and October 2022. The primary objective of the study was to examine the cross-sectional relationship between circulating NFL levels and aging, walking speed, body composition, and metabolomic markers. During the study, the participants underwent a comprehensive assessment lasting approximately 3 h, which involved medical, physiological, and psychological examinations conducted at the Hospital Universitario de Son Espases. The study adhered to the principles of Good Clinical Practice and the ethical guidelines outlined in the Declaration of Helsinki. Ethical approval for the project (IB 4337/20 PI, 4 December 2020) was obtained from the ethics committee of the Balearic Islands. Prior to participation, the participants were fully informed about the study’s objectives, procedures, and associated risks. They provided their informed consent and signed a consent document at each visit.

### 4.2. Physical Performance Test

The assessment of walking speed was conducted using the 4-m walking test at the usual and fast pace. Participants were instructed to wear comfortable clothing and suitable footwear for walking. A distance of 4 m was measured and marked to indicate the starting and ending points on the floor. Prior to the test, we provided a demonstration to familiarize the participants with the procedure. A trial walk was then performed by each participant. They were instructed to walk at their usual or at a fast pace in a 4-m distance on the floor. The timing commenced from the first foot movement and stopped when the participant’s foot made contact with the floor at the end of the walking course.

Muscle strength was evaluated using grip strength torque measurements and 5-times sit-to-stand test. Briefly, grip strength was assessed using a Kern hand dynamometer that was calibrated with known weights and adjusted to ensure hand comfort and proper fit. Participants were instructed to keep their arms in a relaxed and stationary position. Three maximum grip measurements were taken, and the highest value recorded for each hand. During the 5-times sit-to-stand test, participants were seated with their back against the chair’s backrest and performed five consecutive rises from the chair. They were instructed to rise without using their arms and without pausing between each repetition, aiming for the highest speed possible. Each stand was audibly counted to maintain the participant’s orientation. The test concluded once the participant successfully achieved the standing position for the fifth repetition.

### 4.3. Measurements of Body Composition

To assess body composition, we used the Inbody 770, a bioelectrical impedance analysis (BIA) machine (InBody Co., Ltd., Seoul, Republic of Korea). Prior to the test, participants were provided with instructions to ensure accurate measurements. They were required to fast for a minimum of 12 h, dress in lightweight clothing, refrain from wearing any metal objects or electronic medical devices, and avoid drinking water for at least 2 h before the test. During the body composition assessment, participants were positioned on the BIA instrument, standing barefoot and aligning themselves with the back foot electrode. They were instructed to keep their arms away from the sides of their body while holding the hand electrodes. Participants maintained this position quietly for approximately 2 min until the testing was completed.

### 4.4. NFL Serum Levels

Serum levels of NFL were quantified using a commercially available ELISA kit from Quanterix Company^®^ (Billerica, MA, USA). The assay kit demonstrated good reproducibility, with intra-assay and inter-assay coefficients of variation (CV) of 6.6% and 10%, respectively.

### 4.5. Measurement of Plasma Metabolites

Blood samples were collected from the antecubital vein between 07:30 and 08:30 h after an overnight fasting period. Prior to the blood sample collection, participants were instructed to refrain from smoking, engaging in physical activity, or taking medications. The collected blood samples were promptly stored at a temperature of 4 °C and were centrifuged within 4 h. Subsequently, the samples were immediately divided into smaller portions and frozen at a temperature of −80 °C. Plasma metabolites were quantified using liquid chromatography tandem mass spectrometry (LC-MS/MS). The extraction of metabolites and the measurement of their concentrations were performed using the AbsoluteIDQ p180 kit (Biocrates Life Sciences AG, Innsbruck, Austria) in accordance with the manufacturer’s protocol for a 6500 QTrap instrument (AB Sciex, Framingham, MA, USA) coupled to an Agilent 1290 Infinity UHPLC system (Agilent, Santa Clara, CA, USA). Metabolites below the limit of detection were excluded from the analysis.

### 4.6. Statistical Analysis

Descriptive characteristics were summarized as means and standard error of the mean (SEM) or standard deviation (SD) or as numbers and percentages (%). Distributions of population characteristics were examined through histograms and boxplots. One-way analysis of variance (ANOVA), Chi-square tests (χ2), and Fisher’s exact test were used to assess differences across genders for continuous and categorical variables, respectively. Non-normally distributed continuous data were compared using the Mann–Whitney-Wilcoxon test or the Kruskal–Wallis H-test. The relationships between variables were studied using Spearman or Pearson correlations and partial correlations. Multivariate linear regression models were used to test the relationships and potential interactions between various independent variables (muscle quality, age, gender and NFL) and the dependent variable of interest (metabolites). All analyses and plots were performed using RStudio 3.5.3. (R Foundation for Statistical Computing, Vienna, Austria) and IBM SPSS statistics 25 (IBM Corp., Armonk, NY, USA).

## Figures and Tables

**Figure 1 ijms-24-12751-f001:**
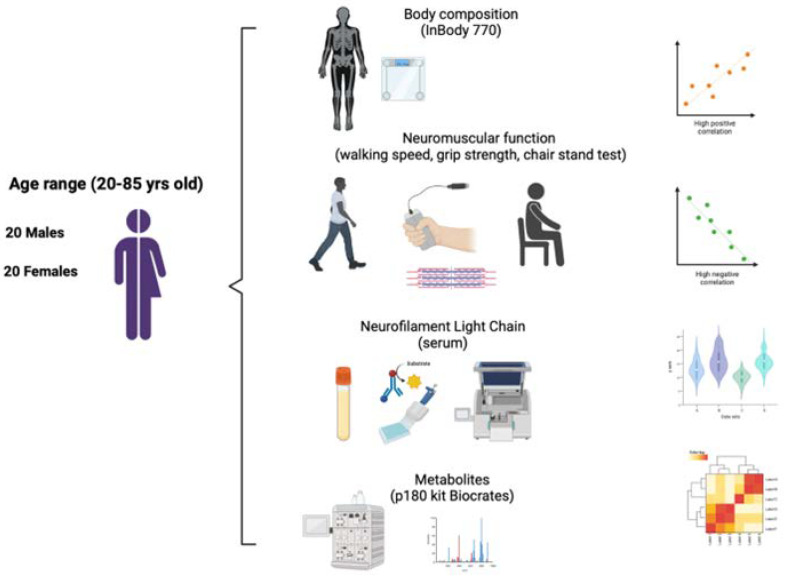
Overview of the study. Figure created with BioRender.com.

**Figure 2 ijms-24-12751-f002:**
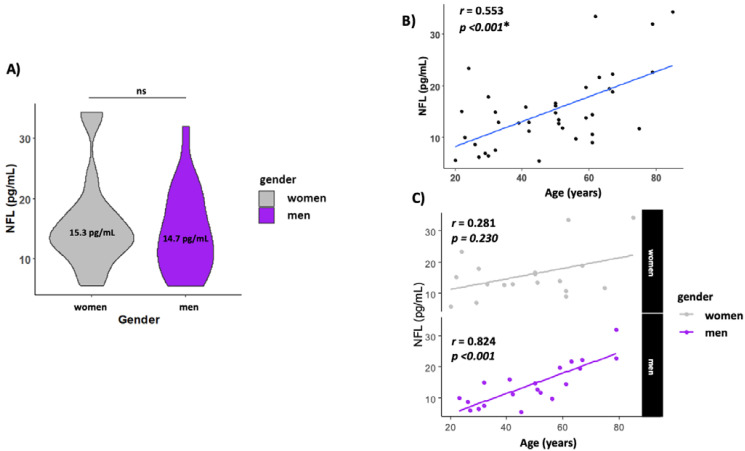
Circulating NFL increase with age. (**A**) Gender NFL levels; (**B**) scatterplot showing the association between age and NFL levels in serum. (**C**) Scatterplot showing the association between age and NFL levels by gender. The line labeled “ns” in Figure 2A means not significant. The * means statistically significant *p* < 0.05.

**Figure 3 ijms-24-12751-f003:**
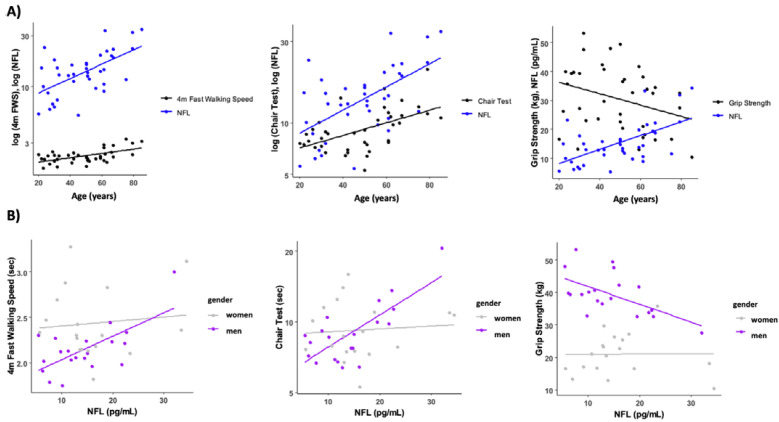
Relationship between circulating NFL, age, and functional performance tests. (**A**) Muscle function test and NFL trajectories with age. (**B**) Association between NFL levels and fast walking speed, time in the chair test and grip strength levels. Some variables have been log transformed to scale variables.

**Figure 4 ijms-24-12751-f004:**
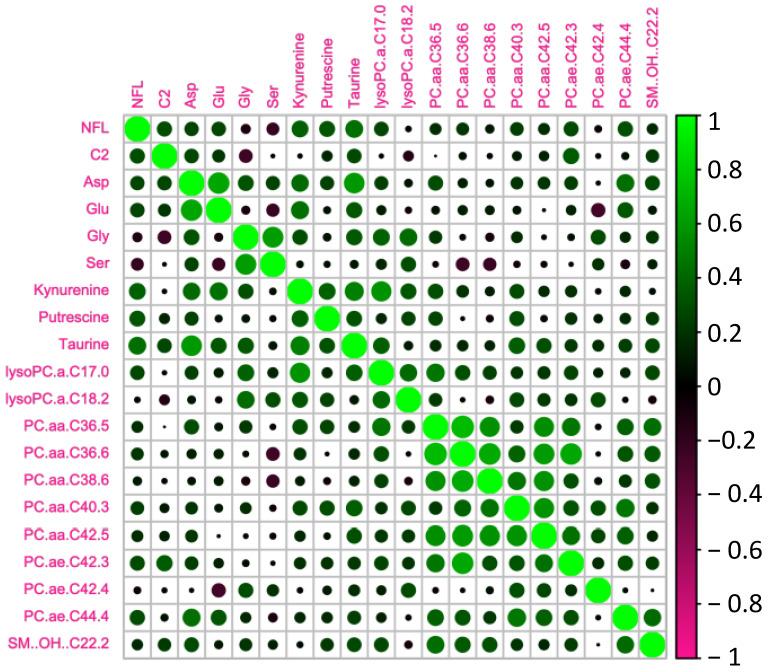
Heatmap of the significant metabolites associated with NFL.

**Table 1 ijms-24-12751-t001:** Characteristics of the 40 participants included in the study.

Participants characteristics				
	**Total**	**Male**	**Female**	** *p* **
**N**	**40**	**20**	**20**	
**Age** (years)	47.8 ± 2.79	48.9 ± 3.83	46.7± 4.15	0.565
**BMI** (kg/m^2^)	23.7 ± 0.54	24.8 ± 0.59	22.5 ± 0.83 *	0.026
**Comorbidities**				
**Hypertension**	3 (7.50%)	2 (10.0%)	1 (5.00%)	0.458
**Asthma**	3 (7.50%)	1 (5.00%)	2 (10.0%)	0.503
**Hypercholesterolemia**	5 (12.5%)	3 (15.0%)	2 (10.0%)	0.516
**Physical performance**				
**4 m gait speed at usual pace** (m/s)	1.17 ± 0.03	1.16 ± 0.03	1.18 ± 0.03	0.632
**4 m gait speed at fast pace** (m/s)	1.81 ± 0.04	1.89 ± 0.05	1.71 ± 0.05 *	0.022
**Grip Strength** (kg)	30.4 ± 1.76	39.4 ± 1.45	21.4 ± 1.42 *	<0.001
**Chair test** (s)	9.39 ± 0.47	9.37 ± 0.74	9.4 ± 0.60	0.980
**NFL**				
**NFL** (pg/mL)	14.7 ± 1.13	14.0 ± 1.54	15.3 ± 1.69	0.561
**Body Composition**				
**Intracellular Water** (**L**)	23.9 ± 0.94	28.6 ± 1.01	19.3 ± 0.57 *	<0.001
**Extracellular area** (**L**)	14.5 ± 0.54	17.3 ± 0.55	11.8 ±0.30 *	<0.001
**Body Fat Mass** (**Kg**)	15.9 ± 1.09	15.6 ± 1.41	16.2 ± 1.69	0.792
**Soft Lean Mass** (**Kg**)	49.4 ±1.91	58.9 ± 2.02	39.9 ± 1.13 *	<0.001
**Fat Free Mass** (**Kg**)	52.4 ± 2.01	62.5 ± 2.14	42.4 ± 1.19 *	<0.001
**Fat percentage** (**%**)	23.4 ± 1.39	19.8 ± 1.58	26.9 ± 2.04 *	<0.001
**Visceral fat area** (**cm^2^**)	74.9 ± 6.24	72.8 ± 7.69	77.1 ± 10.1	0.733
**Waist Hip Ratio**	0.90 ± 0.01	0.91 ± 0.02	0.88 ± 0.02	0.387
**SKM index**	7.20 ± 0.19	8.18 ± 0.19	6.23 ± 0.15 *	<0.001
**50 kH body phase angle**	5.53± 0.13	5.88 ± 0.17	5.16 ±0.15 *	0.003

Values are expressed as mean ± SEM. BMI * Statistically significant *p* < 0.05.

**Table 2 ijms-24-12751-t002:** Correlation coefficients between serum NFL levels and physical performance test, body composition and plasma metabolites in the participants and by gender.

	Total (N = 40)		Males (N = 20)		Females (N = 20)	
	R	*p*	r	*p*	r	*p*
**NFL vs. Physical performance tests**						
**Chair test** (s)	0.509	0.010 *	0.478	0.033 *	0.008	0.975
**Grip strength** (kg)	−0.202	0.211	−0.535	0.015 *	−0.108	0.650
**4 m gait speed at usual pace** (m/s)	−0.223	0.167	−0.460	0.041 *	−0.047	0.843
**4 m gait speed at fast pace** (m/s)	−0.284	0.076	−0.423	0.063	0.131	0.582
**NFL vs. Body composition**						
**Intracellular water**	−0.168	0.300	−0.384	0.095	0.169	0.477
**Extracellular water**	−0.130	0.424	−0.255	0.279	0.220	0.350
**Body fat mass**	0.284	0.075	0.482	0.031 *	0.006	0.980
**Soft Lean Mass**	−0.152	0.349	−0.320	0.168	0.168	0.478
**Fat Free Mass**	−0.153	0.347	−0.331	0.154	0.177	0.456
**Fat Percentage**	0.337	0.034 *	0.496	0.026 *	−0.052	0.828
**Visceral Fat Area**	0.319	0.045 *	0.516	0.020 *	0.112	0.638
**Waist Hip Ratio**	0.155	0.339	0.354	0.126	−0.048	0.840
**50 kHz body phase angle**	−0.406	0.009 *	−0.603	0.005 *	−0.182	0.442
**NFL vs. Metabolomic markers**						
**ASP**	0.371	0.018 *	0.320	0.169	0.446	0.049 *
**GLU**	0.181	0.264	0.445	0.050 *	0.000	1.000
**GLY**	−0.215	0.183	−0.608	0.004 *	0.062	0.796
**SER**	−0.249	0.122	−0.443	0.050 *	−0.055	0.818
**Putrescine**	0.428	0.006 *	0.419	0.066	0.463	0.040 *
**TAU**	0.317	0.046 *	0.293	0.209	0.335	0.148
**KYN**	0.319	0.045 *	0.253	0.283	0.386	0.092
**LysoPC_aa_C17:0**	0.204	0.207	−0.084	0.726	0.492	0.028 *
**LysoPC_aa_C18:2**	−0.111	0.495	−0.458	0.042 *	0.347	0.133
**PC_aa_C36:5**	0.395	0.013 *	0.564	0.010 *	0.214	0.379
**PC_aa_C36:6**	0.309	0.056	0.460	0.041 *	0.096	0.697
**PC_aa_C38:6**	0.191	0.245	0.513	0.021 *	−0.183	0.454
**PC_aa_C42:5**	0.194	0.237	0.462	0.040 *	0.015	0.952
**PC_ae_C38:0**	0.226	0.166	0.477	0.034 *	−0.016	0.949
**PC.aa.C40.3**	0.322	0.046 *	0.299	0.200	0.282	0.241
**PC_ae_C42:3**	0.349	0.030 *	0.110	0.645	0.579	0.009 *
**PC_ae_C44:4**	0.457	0.003 *	0.490	0.028 *	0.447	0.055
**Acetylcarnitine**	0.325	0.041 *	0.335	0.148	0.344	0.137
**OH-Sphingomyelin C22.2**	0.318	0.045 *	0.325	0.128	0.227	0.336

Values are expressed as spearman correlation coefficient * Statistically significant *p* < 0.05

## Data Availability

The data presented in this study are available on request from the corresponding author.

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
