# Peer review of "Circulating Neurofilament Light Chain Levels Increase with Age and Are Associated with Worse Physical Function and Body Composition in Men but Not in Women"

_ijms, 2023, doi:10.3390/ijms241612751_

Round 1

Reviewer 1 Report

Overall the information presented assesses the relationship between age-related changes in the Neuro-filament Light Chain (NFL), a marker of neuronal function. My suggestions are given below

1.      Creatine Kinase (CK) levels is the most frequent and useful laboratory blood test for the diagnosis of neuromuscular diseases. Did the authors check creatine kinase levels in serum/plasma or not? Include creatine kinase levels in the manuscript and its correlation with NFL level.

2.      How does NFL increase in the blood decrease muscular function? What is the molecular mechanism behind this? Does NFL show an effect on axonal degeneration or not? Explain in detail with the help of published articles.

3.      Do these participants have a previous history of neuromuscular/muscular disorders?

4.      Are these participants from similar racial backgrounds or not?

5.      As grip strength is measuring the force applied on the machine, authors need to normalize the data with the corresponding body weight of the participant. It’s not well described in the methodology.

Author Response

Creatine Kinase (CK) levels is the most frequent and useful laboratory blood test for the diagnosis of neuromuscular diseases. Did the authors check creatine kinase levels in serum/plasma or not? Include creatine kinase levels in the manuscript and its correlation with NFL level.

Dear Reviewer, thank you for providing valuable feedback on our manuscript. We appreciate the opportunity to address the concerns raised and clarify our study design and findings.

Regarding the suggestion to include Creatine Kinase (CK) levels in the manuscript, we acknowledge the importance of CK as a frequent and useful laboratory blood test for diagnosing neuromuscular diseases and muscle damage. However, it was not within the scope of our initial research hypothesis to investigate CK levels in serum/plasma as a marker for neuromuscular diseases or muscle damage, mainly because our participants, were totally healthy, active and free of major diseases. Additionally, they were specifically instructed to abstain from exercise 24 hours prior to the study to prevent any elevation of inflammatory markers in the blood that could cofound or impact the results (this has been added to the material and methods, to better clarify the study, section 4, line 334). Our primary aim was to explore the association between Neurofilament Light (NFL) levels and muscle function/mass with aging.  Also, to address the concern of potentially inducing a misunderstanding, we have carefully revised the introduction section, specifically removing the sentence "NFL, a marker of neuronal function..." (line 112), in the discussion (line 259) and in the abstract (line 45). This revision should clarify our research objectives and avoid any confusion regarding the focus of our study.

Also, when there is significant muscle damage (e.g., due to trauma or muscle diseases), the release of CK into the bloodstream may lead to an increase in blood creatinine levels. This is because the injured muscles release CK, and if the kidneys are unable to efficiently clear the increased CK and creatinine, both their levels may rise. Therefore, to try to answer the reviewer suggestion, we evaluated the correlation between circulating creatinine and muscle mass and NFL levels, and found no significant relationship between these markers. As these results did not directly align with the main objectives of the study, we have decided not to include the creatinine data in the main manuscript. However, we have added this data to the supplementary materials for readers interested in further details (Supplemental Figure 1, line 162, results section 2.2).

We hope that these modifications address the concerns raised and provide better clarity on the scope and objectives of our study. We appreciate the feedback and believe that the revised manuscript now better reflects the initial research goals and findings.

How does NFL increase in the blood decrease muscular function? What is the molecular mechanism behind this? Does NFL show an effect on axonal degeneration or not? Explain in detail with the help of published articles.

We appreciate the reviewer's question regarding the relationship between NFL levels in blood and muscular function. While our study focused on evaluating changes in NFL levels with age and their potential association with a decrease in physical function, we acknowledge that the exact molecular mechanisms linking NFL to muscular function are not fully elucidated. However, as walking speed and grip strength are established indicators of neurodegeneration and mortality (Studenski et al., 2011), our findings may suggest that rising levels of NFL during the aging process might be an initial biomarker for certain neurodegenerative processes. [Notably, we observed a tight parallel trajectory between changes in muscle function (characterized by a decline) and NFL levels (characterized by an increase) (see Figure 2). Although we acknowledge that correlation does not imply causation, these observations warrant further investigation into potential underlying mechanisms linking NFL levels and age-related muscle function decline].

As described in the introduction of our paper, NFL is widely recognized as a biomarker of neurodegeneration. Previous studies have indeed reported a correlation between NFL circulating levels and declines in physical function or muscle mass (Monti E et al., 2023; Pratt J et al., 2022). However, we agree that there is no direct evidence showing that NFL in blood directly alters or decreases muscle function 'per se.'

Regarding the molecular mechanisms underlying the association between NFL levels and muscle function, it is plausible that elevated NFL levels could serve as a marker of ongoing axonal damage or degeneration. This axonal damage may result in the loss of muscle function and strength over time. Additionally, NFL may have direct effects on muscle cells or neuromuscular junctions, although more research is needed in this field.

We recognize the importance of further investigating the molecular pathways connecting NFL to muscle function. Longitudinal studies, will be crucial to gaining a comprehensive understanding of the relationship between NFL and muscle function. With our cross sectional design we can't answer this question. 

Do these participants have a previous history of neuromuscular/muscular disorders?

All participants are healthy and any participants in the study have neuromuscular or muscular disorder. This information has been clarified in the methods section (4.1.Study subjects, line 314)

Are these participants from similar racial backgrounds or not?

All participants in the study are Caucasians. This information has been added to the manuscript in the results section (2.1. Participants characteristics, line 135).

As grip strength is measuring the force applied on the machine, authors need to normalize the data with the corresponding body weight of the participant. It’s not well described in the methodology.

Following the reviewer’s recommendation, we have normalized the values of grip strength and chair test with body weight. This data has been added as Supplemental Figure 2 as well as in the result section 2.3, Line 181.

Reviewer 2 Report

The manuscript by Capo et al. investigates a potential association between the nervous/muscular system and age by circulating serum NFL.

The manuscript is correlative and lacks functional aspects of this connection. The experimental point of view is interesting but lacks significance. The experimental plan is well executed, and the data are not very clear and well-written.

The table and statistical analysis are showed not clear, and a reorganization is suggested.

The methods description is correct, and obviously, in the discussion are evidenced many limits of these descriptive results, but this is not due to the authors but to the limited reports of the research theme.

This work is a pilot study.

The literature cited is appropriate.

More molecular and cellular details are requested, to affirm the NfL reported observations.

This work shows the reader a glue composition of independent experimental parts, lacking the important observation about the relationship of Nfl measurement and aminoacid/metabolites as a key to interpreting muscle weakness.

Text editing is suggested.

The manuscript may be not accepted for publication.

Author Response

The manuscript by Capo et al. investigates a potential association between the nervous/muscular system and age by circulating serum NFL. The manuscript is correlative and lacks functional aspects of this connection. The experimental point of view is interesting but lacks significance. The experimental plan is well executed, and the data are not very clear and well-written. The table and statistical analysis are showed not clear, and a reorganization is suggested. The methods description is correct, and obviously, in the discussion are evidenced many limits of these descriptive results, but this is not due to the authors but to the limited reports of the research theme. This work is a pilot study. The literature cited is appropriate.

We sincerely thank the reviewer for its thoughtful evaluation of our manuscript. We carefully considered each comment and appreciate the constructive feedback provided.  We acknowledge that our study is correlative in nature (and cross sectional), and we agree that functional or cellular mechanisms or aspects of this connection could be missing but at the same time we also believe it could provide valuable insights. As the reviewer rightly pointed out, our experimental plan was designed to explore potential associations and serve as a pilot study to shed light on this research theme. We also appreciate the reviewer's recognition of the limitations inherent in our descriptive results. Indeed, given the complexity of the research theme and the pilot nature of our study, we agree that there are inherent limitations. However, we have thoroughly discussed these limitations in our manuscript to provide a comprehensive understanding of the implications of our findings and we have improved our manuscript and addressed some of the concerns along the manuscript.

More molecular and cellular details are requested, to affirm the NfL reported observations.

We appreciate the reviewer's comment about the molecular and cellular aspects of our study. As mentioned previously in response to reviewer 1, our primary focus was to evaluate changes in NFL levels with age and their potential association with a decrease in physical function. With our study we wanted to explore if circulating NFL could be a potential early marker of a decline in muscle function and or/age-associated neurodegenerative processes.

We agree that the exact molecular mechanisms linking NFL to muscular function require further investigation, and we acknowledge that our study was not specifically designed to analyze these detailed molecular mechanisms. While our study aimed to assess the relationship between NFL circulating plasma levels and well-known proxy markers of muscle degeneration, such as grip strength, walking speed, and the chair test, we acknowledge that it is essential to study deeper into the underlying molecular and cellular processes.

Again, we greatly appreciate this valuable comment, and we will certainly consider it for future studies. For instance, to address this question in the future, effectively, an ideal model would involve utilizing C. elegans, for instance, for mechanistic investigations. In humans, studying participants in the initial stages of PD or AD, or following our participants longitudinally, could help to gain deeper insights into the role of NFL as potential marker of early stage of age-associated neurogenerative processes. Therefore, in the future, this could help us to study the potential of NFL as a diagnostic tool for identifying these processes in their early stages.

This work shows the reader a glue composition of independent experimental parts, lacking the important observation about the relationship of NFL measurement and aminoacid/metabolites as a key to interpreting muscle weakness.

We acknowledge the importance of providing a comprehensive interpretation of the relationship between NFL measurements and amino acids/metabolites, particularly in the context of muscle weakness. While we did analyze circulating metabolites using the AbsoluteIDQ® p180 kit (Biocrates) and presented correlations between NFL levels and the most significant metabolomic markers in Table 2 and Figure 4, we acknowledge the reviewer's valid point about the potential for additional insights into the specific interactions between NFL and amino acids/metabolites, which could enhance the understanding of muscle weakness mechanisms.  However, it is important to note that the primary scope of our study was not focused on investigating the detailed molecular interactions between NFL and these metabolites with muscle weakness. The objective of our analysis was to explore whether NFL was associated with classical metabolomic markers that are known to change with aging or that could play a role in muscle function. Specifically, some metabolites have been reported in the literature to be associated with a decline in muscle function, such as LysoPC 18:2, putrescine, kynurenine, BCAA amino acids, or aromatic amino acids. Given the relevance of these metabolites and their potential implications in muscle function, we sought to examine their associations with NFL. But we certainly agree that investigating the specific molecular interactions between NFL, muscle weakness and the metabolites included in the study would undoubtedly be a valuable point. We appreciate the reviewer's thoughtful consideration of this aspect. Therefore, we have added additional analysis, adjusting for muscle quality, a proxy marker of muscle weakness. (results section, 2.4, line 227 “Considering that the correlations between NFL and these metabolites could be influenced by muscle quality, we conducted multivariate linear regression analyses adjusting for age, gender, and muscle quality. Consequently, some of the associations observed initially disappeared; however, others, including Acetylcarnitine (C2) and PC_ae_C40.3, remained significant. The complete set of beta coefficients and corresponding p-values for NFL and the metabolites can be found in Supplemental Table 1”. We have also added some sentences in the discussion section (line 312).

Text editing is suggested. Thank you for this comment. We have taken this comment in consideration.

Round 2

Reviewer 2 Report

The manuscript shows several limitations and a lack of novelty and functional interpretation.